

# Omics approaches to unravel insecticide resistance mechanism in *Bemisia tabaci* (Gennadius) (Hemiptera: Aleyrodidae)

Muhammad Aqil Fitri Rosli[1], Sharifah Nabihah Syed Jaafar[2], Kamalrul Azlan Azizan[1], Salmah Yaakop[3] and Wan Mohd Aizat[1]

[1] Institute of Systems Biology (INBIOSIS), Universiti Kebangsaan Malaysia, Bangi, Selangor, Malaysia
[2] Department of Applied Physics, Faculty of Science and Technology, Universiti Kebangsaan Malaysia, Bangi, Selangor, Malaysia
[3] Centre for Insect Systematics, Faculty of Science and Technology, Universiti Kebangsaan Malaysia, Bangi, Selangor, Malaysia

## ABSTRACT

*Bemisia tabaci* (Gennadius) whitefly (BtWf) is an invasive pest that has already spread worldwide and caused major crop losses. Numerous strategies have been implemented to control their infestation, including the use of insecticides. However, prolonged insecticide exposures have evolved BtWf to resist these chemicals. Such resistance mechanism is known to be regulated at the molecular level and systems biology omics approaches could shed some light on understanding this regulation wholistically. In this review, we discuss the use of various omics techniques (genomics, transcriptomics, proteomics, and metabolomics) to unravel the mechanism of insecticide resistance in BtWf. We summarize key genes, enzymes, and metabolic regulation that are associated with the resistance mechanism and review their impact on BtWf resistance. Evidently, key enzymes involved in the detoxification system such as cytochrome P450 (CYP), glutathione S-transferases (GST), carboxylesterases (COE), UDP-glucuronosyltransferases (UGT), and ATP binding cassette transporters (ABC) family played key roles in the resistance. These genes/proteins can then serve as the foundation for other targeted techniques, such as gene silencing techniques using RNA interference and CRISPR. In the future, such techniques will be useful to knock down detoxifying genes and crucial neutralizing enzymes involved in the resistance mechanism, which could lead to solutions for coping against BtWf infestation.

## INTRODUCTION

*Bemisia tabaci* (Gennadius) (BtWf) or silverleaf whitefly is globally known as one of the most devastating pests due to their invasive nature. Initially, BtWf was identified to originate from the subtropical regions and tend to inhabit a temperate environment (*Perring et al., 2018*). Subsequently, this species has been distributed worldwide because of its polyphagous diet and rapid breeding (*Abubakar et al., 2022*; *Ghahari et al., 2009*).

Corresponding author
Wan Mohd Aizat,
wma@ukm.edu.my

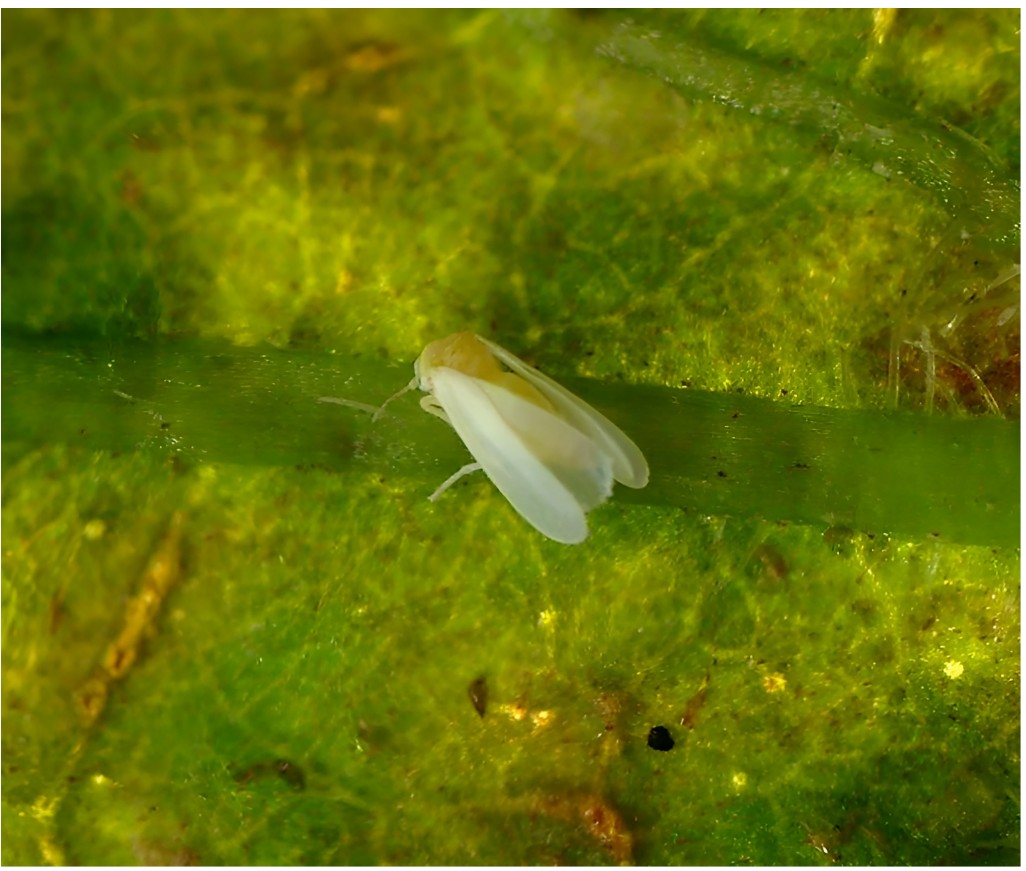

**Figure 1  A close-up image of BtWf unique modified mouthparts from a plant foliage.** Photo credit: Muhammad Aqil Fitri Rosli. The adult BtWf was observed with a trinocular stereoscopic microscope SZ61 (Olympus) attached with a digital camera. This camera was linked and controlled *via* ToupView (version 3.0).                                                

Among 40 morphologically indistinguishable cryptic species of BtWf, Middle East-Asia Minor one species (Biotype B or MEAM1), and Mediterranean species (Biotype Q or MED) are the most invasive and prominent genetic groups due to their global invasion (*Abubakar et al., 2022*; *Horowitz et al., 2020*; *Li et al., 2021*; *Wang et al., 2011*; *Watanabe et al., 2019*). Whiteflies use their unique modified mouthparts to pierce (Fig. 1) and suck the sap from plant foliage (*Onstad & Knolhoff, 2022*). Such feeding behavior leads to honeydew accumulation on the foliage surface, inviting mold growth and thereby affecting the plant's photosynthetic efficiency (*Farina et al., 2022*; *Perring et al., 2018*). This damage gradually weakens infected plants, reducing their yield and eventually killing the plants (*Farina et al., 2022*; *Perring et al., 2018*). Furthermore, according to *Sani et al. (2020)*, BtWf can vector and transmit more than 457 viruses, earning it the title of "super-vector" (*Ghosh, 2021*; *Gilbertson et al., 2015*). Begomovirus has been the most actively transmitted by BtWf, causing 20% to 100% crop yield losses amounting to billions of USD (*Gangwar & Gangwar, 2018*; *Sani et al., 2020*). Consequently, the BtWf invasion could threaten global farming and agricultural industries, particularly for Solanaceae species such as chilies and tomatoes (*Farina et al., 2022*; *McKenzie et al., 2014*).
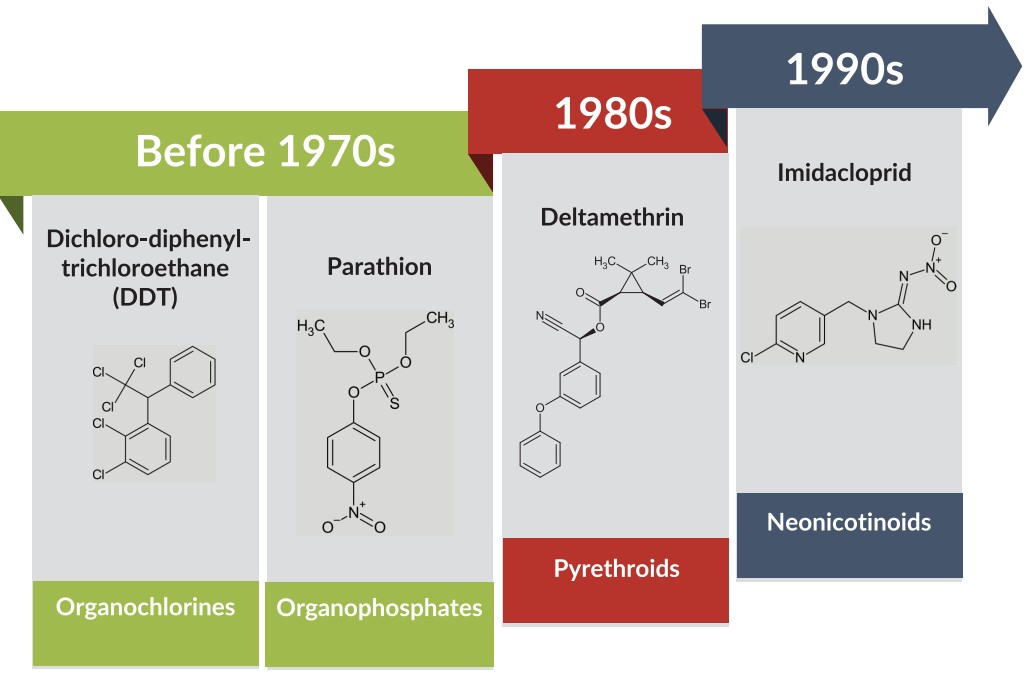

**Figure 2** **BtWf insecticide resistance status over time across various classes of insecticides.** Data source: *Naveen et al. (2017)* and *Siddiqui et al. (2023)* Template credit: Koleksi Template PowerPoint GayaPro v3.0 (Light). Chemical structure drawing tool: ChemSketch (Freeware) 2023.2.4 by ACD/Labs.

Even though several alternatives have been introduced in pest management, such as biological control measures, the use of chemical insecticides remains the most efficient and cost-effective method (*Guo et al., 2017*; *Siddiqui et al., 2023*). However, BtWf has evolved resistance to a wide spectrum of insecticides because of overused and prolonged insecticide exposure (*Horowitz et al., 2020*; *Siddiqui et al., 2023*). During the late 1970s and early 1980s, insecticides such as pyrethroids have been increasingly used to replace organophosphates and organochlorines due to the BtWf resistance (*Head & Savinelli, 2008*). In response, neonicotinoids such as thiamethoxam and imidacloprid, a new class of insecticide that targets the insect nervous system (*Naveen et al., 2017*) was introduced in the late 1990s to further curb the pest infestation (Fig. 2).

Unfortunately, the *Bemisia* family has been reported to be resistant in approximately 650 cases which involve over 60 different active components, including thiamethoxam and imidacloprid (*Horowitz et al., 2020*; *Naveen et al., 2017*), and this progressive insecticide resistance in BtWf has raised concern worldwide. According to *Khan et al. (2020)*, insects such as BtWf can resist at the metabolic level (metabolic resistance) by expressing enzymes to break down and metabolize insecticides before reaching or binding to the target molecular site. These enzymes are usually involved in the detoxification of xenobiotics, which are comprised of cytochrome P450s (CYP), carboxylesterases (COE), glutathione S-transferases (GST), and ABC transporters (ABC) (*Chen et al., 2016*; *Wang et al., 2023*; *Yang et al., 2013a*). Another known resistance mechanism is called target-site resistance caused by modified (mutated) insecticide binding sites in the host organism

**Table 1 Recent studies on multi-omics analyses related to insecticide resistance in *B. tabaci* (BtWf).**

| Experimental design | Technologies | Findings | Impacts on BtWf | Source |
|---|---|---|---|---|
| **Genomics and transcriptomics** | | | | |
| Comparative transcriptomic of different strains of MED BtWf on the allelic variation of *CYP6CM1* | Illumina sequencing (HiSeq 2500) | Both strains have elevated expression of 8 COE, 5 GST, and 39 CYP. *CYP402B2, CYP303A1, CYP380G1*, and *CYP6CM1* showed significant overexpression. | Detoxification of neonicotinoids imidacloprid and clothianidin | *Pym et al. (2023)* |
| Genome-wide and expression analysis of the CYP gene family of MED BtWf | PacBio | 24 CYP genes were identified and classified into four families: CYP4, CYP6, CYP9, and CYP301-318, with nine genes from the CYP4 and CYP6 families were highly upregulated due to imidacloprid exposure. | Imidacloprid resistance | *Qin et al. (2023)* |
| Identification of insecticide resistance markers *via* comparative genomics analysis of different BtWf lines | Illumina sequencing (Hi-Seq 2500) | 57 CYP, 14 COE, 28 UGT, 2 GST, and 16 ABC transporters with non-synonymous SNPs were identified | Detoxification and insecticide resistance | *Wang et al. (2023)* |
| Transcriptomics profile of imidacloprid (IMI), acetamiprid (ACE), and thiamethoxam (THI) treated MEAM1 BtWf | Illumina sequencing (NovaSeq) | IMI strain had DEG mostly CYP and cuticle proteins. While ACE and THI strains had CYP, GST, nAChR, mAChR, HSP, and COE. | IMI strain exhibited the highest resistance, followed by the ACE strain and then the THI strain | *Zhou et al. (2022)* |
| Amplicon sequencing on different strains of BtWf in China | Amplicon & Sanger sequencing | *L925I* and *T929V* mutations in Voltage-gated sodium channels (VGSC) | Pyrethroid resistance | *Wei et al. (2021)* |
| RNA sequencing of five selected strains of MEAM1 and MED biotype BtWf with different resistance levels | Illumina sequencing NextSeq 500 | A2083V mutation in the acetyl-CoA carboxylase (ACC) gene, which confers high resistance to these insecticides | Keto-enol resistance | *Lueke et al. (2020)* |
| Comparative transcriptomics analysis between MED and GH BtWf | Illumina sequencing (Hi-Seq 2000/2500) | MED has a GST gene and 4 CYP genes including *CYP4C64*. | Imidacloprid resistance | *Wang et al. (2020b)* |
| Genome-wide analysis of UGTs in MEAM1 biotype BtWf | Genome sequence data from *Chen et al. (2016)* were obtained using Illumina sequencing (Hi-Seq 2500) and PacBio. Transcriptome data from *Xie et al. (2014)* were acquired using Illumina sequencing (Genome AnalyzerIIx). | 76 UGT genes were identified in the MEAM1 biotype BtWf genome | The detoxification roles of UGTs enhance BtWf's ability to adapt to a variety of host plants | *Guo et al. (2020b)* |
| Genome-wide analysis of COEs in MED biotype BtWf | Datasets from *Xie et al. (2017)* using Illumina sequencing (Hi-Seq 2000) | 42 putative COEs in different functional categories were identified in MED biotype BtWf with six COEs were highly expressed when exposed to imidacloprid. | Detoxification of xenobiotics and insecticides | *Xia et al. (2019)* |

| Experimental design | Technologies | Findings | Impacts on BtWf | Source |
|---|---|---|---|---|
| Genome-wide analysis of ABC transporters in MED biotype BtWf | Illumina sequencing (Hi-Seq 4000) | 55 ABC transporter genes were discovered and 80% of the ABCG subfamily were highly up-regulated in imidacloprid-treated condition | Imidacloprid resistance | *Tian et al. (2017)* |
| Assembly of BtWF MEAM1 draft genome sequence *via* genomics and transcriptomics platform | Illumina sequencing (Hi-Seq 2500) and PacBio | Identified 130 CYP genes, 81 UGT genes, 22 GST genes, 50 ABC genes, and 51 COE genes | Detoxification and insecticide resistance | *Chen et al. (2016)* |
| Transcriptomics analysis of neonicotinoid resistance in MED populations | Illumina sequencing (Hi-Seq 2000) | Five CYP genes, including *CYP6CM1*, *CYP303*, and *CYP6CX3* were linked to neonicotinoids resistance | Imidacloprid and acetamiprid resistance | *Ilias et al. (2015)* |
| Comparative transcriptomics analysis of MED and MEAM1 BtWf guts | Illumina sequencing (Hi-Seq 2000) | Three CYP genes and 1 GST gene were highly up regulated in MED gut compared to MEAM1 gut | High insecticide resistance in MED | *Ye et al. (2014)* |
| Transcriptomics profiling of different developmental stages of thiamethoxam-resistant and susceptible strains BtWf | Microarray | Thiamethoxam-resistant strain (TH-R) consists of CYP genes (*CYP6CM1*), GST genes, COE genes, and ABC genes | Neonicotinoid resistance | *Yang et al. (2013a)* |
| **Proteomics** | | | | |
| Proteomic analysis of MEAM1 BtWf to identify functional genes for pest control | 2-DGE coupled with MALDI TOF/TOF | GST and COE2 protein spots were identified | Detoxification ability | *Mishra et al. (2016)* |
| Comparative proteomics analysis of thiamethoxam-resistant and susceptible resistant BtWf | LC-MS/MS (LTQ-Orbitrap-Velos) and enzymatic assays | 3 DEP involved in insecticide detoxification, namely GST, UGT, and glucosyl/glucuronosyl transferase | Thiamethoxam detoxification and resistance | *Yang et al. (2013b)* |
| Comparative proteomics analysis of MEAM1 and MED BtWf | 2DGE coupled with LC-MS/MS (QSTAR) | Overexpression of COE2 can confer chemical defence in MEAM1 | Pyrethroid and organophosphate resistance | *Kang et al. (2012)* |

**Note:**
Abbreviations: 2-D gel electrophoresis (2DGE), ATP binding cassette transporters (ABC), carboxylesterases (COE), cytochrome P450s (CYP), differentially expressed proteins (DEP), Greenhouse whitefly (GH), glutathione S-transferases (GST), Middle East-Asia Minor 1 species (MEAM1), Mediterranean species (MED), and UDP-glucuronosyltransferases (UGT).

(*Horowitz et al., 2020*). These two resistance mechanisms suggest that regulation at the molecular level is key to unraveling the underlying BtWf resistance to develop new effective eradication strategies in the future.

High-throughput systems biology approaches such as genomics, transcriptomics, proteomics, and metabolomics offer systems-level, wholistic molecular insights into any biological entity including BtWf and its response to insecticides (*Friboulet & Thomas, 2005*; *Ullah et al., 2022*). However, there are yet no recent reviews on this topic and

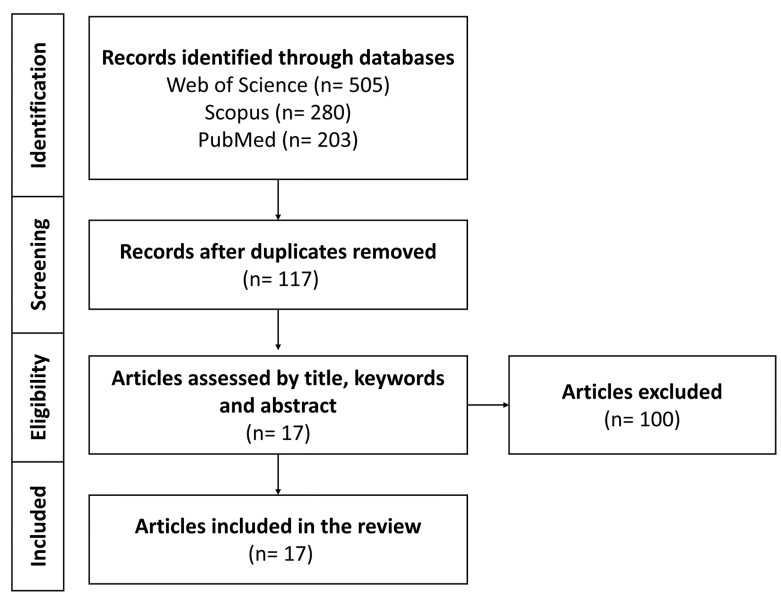

**Figure 3** **A PRISMA model on multi-omics platforms application in understanding BtWf insecticide resistance research.** The model encompassing of database searches with keywords "*Bemisia tabaci*" AND "insecticide resistance" to identify relevant articles and establishing specific inclusion and exclusion criteria.                                                 

therefore we have gathered studies reported in the last 12 years (up to 2023) to shed some light on the applications of omics in understanding BtWf resistance to insecticide (Table 1). Therefore, this review aims to elucidate the mechanisms behind BtWf insecticide resistance, focusing on their detoxification system as revealed by recent multi-omics analyses. Furthermore, we detail the stages of the insect resistance mechanisms and describe current trend strategies such as RNAi-and CRISPR-based gene editing to combat BtWf.

## SURVEY METHODOLOGY

We compiled scholarly articles from a diverse array of sources, encompassing esteemed databases such as Web of Science, Scopus, and PubMed. This process involved executing meticulous searches employing specific keywords and combinations, comprised of ("*Bemisia tabaci*" OR "*B. tabaci*") AND (insecticide OR pesticide OR resistance) AND (omic* OR CRISPR OR RNAi) within the search field. Furthermore, our selection criteria were stringent and focused on recent publications spanning the period from 2011 to 2023. Figure 3 displayed a PRISMA model outlining the criteria for the inclusion and exclusion of research articles in the review focused on exploring the applications of multi-omics platforms in comprehending BtWf insecticide resistance. These chosen articles were aimed to furnish current emerging research trends in the application of multi-omics technology to unveil insecticide resistance in BtWf and the contemporary strategies deployed to manage their infestations. Therefore, this comprehensive review could cater the individuals who are interested in agriculture, pest management strategies, sustainable crop

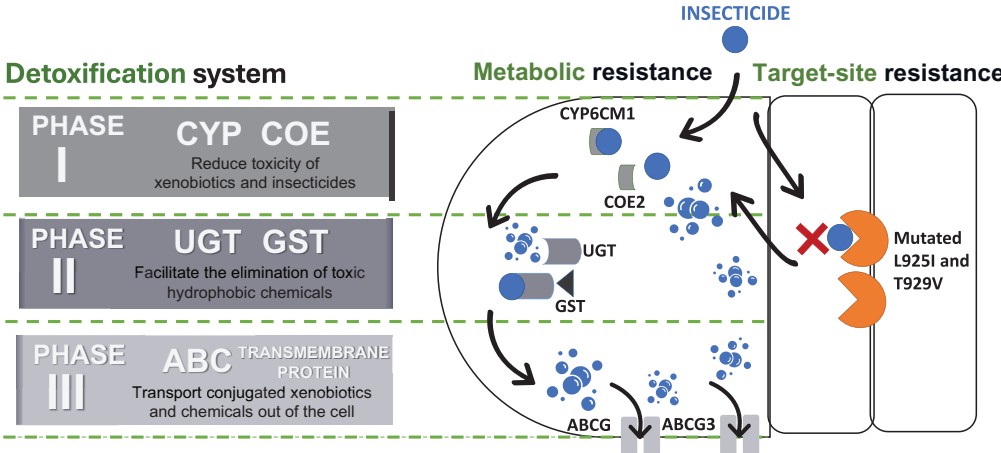

**Figure 4 Detoxification systems in insects for resisting insecticides.** ATP binding cassette transporters (ABC), carboxylesterases (COE), cytochrome P450s (CYP), glutathione S-transferases (GST), and UDP-glucuronosyltransferases (UGT).               

protection, and actively engaged in insecticide resistance mechanism research, particularly in BtWf.

## MECHANISM OF INSECTICIDE RESISTANCE IN BtWf

The detoxification systems are important aspects of the metabolic resistance mechanism in insects during invasion (*Khan et al., 2020*). According to *Gao et al. (2022)* and *Yang et al. (2013b)*, detoxification systems in insects can be classified into three phases and each of these phases consists of a superfamily of enzymes/proteins (Fig. 4).

Phase I involves detoxification enzymes, including cytochrome P450 monooxygenase (CYP) and carboxylesterases (COE) (*Lin, Yang & Yao, 2022*). These enzymes work to mitigate the toxicity of a wide range of xenobiotics by reducing the biological activity of various endogenous and exogenous toxic substances through processes like oxidation, hydrolysis, and reduction (*Abubakar et al., 2022*; *Eakteiman et al., 2018*; *Lin, Yang & Yao, 2022*; *Yang et al., 2013b*) (Table 1). CYP is a heme-containing monooxygenases superfamily known for their versatility and capability to perform mixed-function oxidation, enabling the enzyme to insert an oxygen atom into the unreactive C–H bonds of organic molecules, while reducing another oxygen atom to water (*Chiliza, Martínez-Oyanedel & Syed, 2020*; *Lu, Song & Zeng, 2021*; *Podgorski et al., 2022*; *Yadav et al., 2023*). This capability allows CYPs to play a critical role in breaking down endogenous and exogenous compounds, including defense against synthetic insecticides, due to their genetic diversity and broad substrate specificity (*Feyereisen, 2006*; *Lu, Song & Zeng, 2021*; *Nauen et al., 2022*; *Yang et al., 2013a*). Meanwhile, COE has the advantage to hydrolyze the ester bond, that leads to the disruption of a significant bond that preserve and maintain the structure of certain insecticides, such as pyrethroids, organophosphates, carbamates, and even neonicotinoids (*Khan et al., 2020*). Therefore, both of the enzyme families were frequently targeted and constitutively overexpressed to justify their contribution to

detoxifying xenobiotics and conferring insecticide resistance (*Lu, Song & Zeng, 2021*; *Xia et al., 2019*).

Phase II reactions involve the process of conjugation of xenobiotics and Phase I products into hydrophilic compounds, which are more water-soluble substances such as sugars, sulphates, phosphates, amino acids, or glutathione (*Koirala, Moural & Zhu, 2022*; *Yu, 2008*). This conjugated product exhibits high polarity compared to its precursor compound, which makes it less toxic and easily excreted from insects (*Lin, Yang & Yao, 2022*; *Yang et al., 2013b*; *Yu, 2008*). UDP-glucuronyl transferases (UGTs) and glutathione transferases (GSTs) are the enzyme superfamilies involved in these reactions (*Koirala, Moural & Zhu, 2022*; *Yang et al., 2013b*). For instance, UGTs enhance the water solubility of xenobiotics and insecticides by catalysing the attachment of glycosyl groups from activated sugar donors to a broad range of hydrophobic toxic substances (*Li et al., 2018*). Whilst GSTs catalyse the bonding of reduced glutathione (GSH) to electrophilic toxic substances produced during Phase I reactions (*Meng et al., 2022*). Furthermore, the GSTs are not only able to metabolize toxic substances but also help to eliminate oxidative free radicals from insecticide exposure (*He et al., 2018*; *Meng et al., 2022*; *Wang et al., 2022*). Evidently, several studies have shown that Phase II enzymes can conjugate major neonicotinoid insecticides such as thiamethoxam and imidacloprid in BtWf (*Wang et al., 2021*; *Yang et al., 2013b*) (Table 1).

Then, the detoxification process advances to Phase III, the final phase, which involves the elimination or removal of conjugated xenobiotics and metabolites from the cell (*Yang et al., 2013b*). While some metabolites may have been made less toxic through the reactions in Phases I or II, they can also be directly excreted without undergoing further chemical alteration (*Hilliou et al., 2021*). This phase predominantly relies on ATP-binding cassette (ABC) and other transmembrane transporters. ABC transporters are proteins that have the function of transporting various types of substances, such as amino acids, peptides, sugars, and hydrophobic compounds across the membrane (*Campbell et al., 2023*). These ABC proteins are active transporters that rely on ATP for transporting substrate across the lipid membranes (*Amezian, Nauen & Van Leeuwen, 2024*; *Denecke et al., 2021*). The superfamily of ABC transporters in insects can be categorized into eight subfamilies (ABCA to ABCH) with diverse ATP-binding sites (*Merzendorfer, 2014*; *Wu et al., 2019*). Several members of the insect ABC transporter superfamily (ABCA, ABCB, ABCC, and ABCG) significantly contribute to insecticide resistance by actively preventing the accumulation of insecticides and xenobiotics within their cells (*Denecke et al., 2021*; *Wu et al., 2019*).

## OMICS ANALYSES REVEAL KEY REGULATORY MECHANISMS OF BtWf RESISTANCE

### Genomics and transcriptomics

Advances in sequencing platforms have greatly expanded the prospect of genome and transcriptome research. The evolution of sequencing technologies, from automated DNA sequencing in 1986 to Next-Generation Sequencing (NGS) in 2000, has established a
standard platform for genomic variants identification (*Aizat, Goh & Baharum, 2018*; *Gasperskaja & Kučinskas, 2017*). Genomics involves a comprehensive analysis of an organism's entire genome, utilizing DNA sequencing, recombinant DNA technologies, and bioinformatics to analyse the genome structure (structural genomics and gene mapping) and biological functions (comparative and functional genomics) (*Abdullah-Zawawi et al., 2022*; *Kumar Yadav et al., 2022*). Meanwhile, transcriptomics focuses on analysing RNA transcripts to understand gene expression patterns, which leads to the discovery of splice isoforms, fusion transcripts, and ncRNAs (non-coding RNA) (*Casamassimi & Ciccodicola, 2019*; *Hampel et al., 2021*). Transcriptomics research that focuses on insecticide exposure and the way it alters gene expression patterns may aid in comprehending the molecular mechanisms behind insecticide resistance (*Tang et al., 2014*).

Both omics analyses enable the acquisition of qualitative and quantitative data on gene or transcript expression levels across various conditions and treatments (*Hampel et al., 2021*; *Ullah et al., 2022*). These advancements in genomics and transcriptomics, such as their read lengths, throughput, and accuracy, offer greater insights into downstream molecular expression and cellular processes (*Jamil et al., 2020*; *van Dijk et al., 2023*). Additionally, the emergence of advanced long-read sequencing technologies like Oxford Nanopore and Pacific Biosciences (PacBio) provides more cost-effective solutions with read lengths exceeding 25 kb, further enhancing the discovery of mechanisms in BtWf (*Abdullah-Zawawi et al., 2022*; *Amarasinghe et al., 2020*; *Hotaling et al., 2021*; *Jamil et al., 2020*; *Shi et al., 2023*).

Most genomics and transcriptomics analyses in BtWf utilize Next Gene Sequencing (NGS) technologies such as PacBio Single-molecule real-time (SMRT) and Illumina Hi-Seq as well as the conventional microarray platform (Table 1). Most of the recent studies compared different variants of BtWf, to investigate their invasiveness and susceptibility to insecticides (*Wang et al., 2020b*; *Wei et al., 2021*; *Yang et al., 2013a*; *Ye et al., 2014*). These studies hypothesized that the invasive variant of BtWf has higher resistance towards insecticides and has a broad range of host preferences (*Wang et al., 2020b*). Therefore, several biological factors, such as genes and metabolism that contribute to BtWf survival advantage could be determined among the variants.

Current studies of Phase I genes in the two most prevalent BtWf biotypes identified 130 CYP and 51 COE genes in the MEAM1 biotype *via* an integrated DNA and RNA sequencing analysis (*Chen et al., 2016*). In contrast, only 24 CYP and 42 COE genes were identified in the MED biotype from two different targeted genome-wide analyses (*Qin et al., 2023*; *Xia et al., 2019*). Yet, in a comparative transcriptomics analysis of the gut of these two BtWf biotypes, three CYPs were found to be highly upregulated in MED biotype, but none showed upregulation in the MEAM1 biotype (*Ye et al., 2014*). This suggests that the MED biotype may possess greater resistance to insecticides and environmental stresses (*Ye et al., 2014*). A similar scenario was observed in another comparative transcriptomics analysis between MED biotype BtWf with a greenhouse whitefly (*Trialeurodes vaporariorum* Westwood), which showed the over-expression of four CYP genes provided
a survival advantage to the MED biotype, enhancing its tolerance to imidacloprid compared to the greenhouse whitefly (*Wang et al., 2020b*).

Subsequently, a comparative genomics analysis that focused on non-synonymous SNPs on different insecticide susceptibility of MED biotype BtWf lines manages to discover 57 CYP and 14 COE genes as Phase I insecticide resistance markers (*Wang et al., 2023*). Most of the reported CYPs could be classified into four families, namely *CYP4, CYP6, CYP9*, and *CYP301-308* (*Ilias et al., 2015*; *Pym et al., 2023*; *Qin et al., 2023*; *Zhou et al., 2022*). Notably, *CYP6CM1* genes are the most frequently associated with imidacloprid resistance and involved in the detoxification of other neonicotinoids, such as clothianidin and thiacloprid in both MEAM1 and MED biotypes BtWf (*Ilias et al., 2015*; *Pym et al., 2023*; *Qin et al., 2023*; *Yang et al., 2013b*). Furthermore, COEs are another main pillar that works alongside CYPs to detoxify ester-containing xenobiotics and insecticides (*Lin, Yang & Yao, 2022*; *Xuechun, Ming & Nannan, 2018*). According to *Xia et al. (2019)*, six COEs were highly expressed when exposed to imidacloprid, but only *BTbe5* and *BTjhe2* were found to facilitate MED biotype BtWf's resistance to imidacloprid.

Among Phase II detoxification system, UGTs have a higher number of genes identified compared to GSTs in both genomics and transcriptomics analyses of BtWf (Table 1) (*Chen et al., 2016*; *Guo et al., 2020b*; *Wang et al., 2023*). This finding explains BtWf's invasive and prevalent ability to infest numerous crops, as UGTs enhance BtWf's adaptation to various host plants (*Guo et al., 2020b*). Currently, more Phase II genes were reported in the MEAM1 biotype of BtWf compared to the MED biotype BtWf (Table 1), through genomics and transcriptomics integration as reported by *Chen et al. (2016)*. Nonetheless, in a comparative transcriptomics analysis between MEAM1 and MED biotype BtWf, a highly upregulated GST gene was detected in the gut of the MED biotype, with no detection in the MEAM1 biotype (*Ye et al., 2014*). Additionally, another comparative transcriptomics study also revealed the over-expression of a GST gene in the MED biotype BtWf when exposed to imidacloprid, compared to the greenhouse whitefly (*Trialeurodes vaporariorum* Westwood) (*Wang et al., 2020b*). These analyses highlight the role of GST, a Phase II detoxification enzyme in conferring tolerance to insecticides and environmental stresses (*Wang et al., 2020b*; *Ye et al., 2014*).

As for Phase III, genome-wide analyses in two different insecticide resistance studies (Table 1) discovered abundant numbers of ABC transporters in both invasive BtWf biotypes, MEAM1 (50 ABC transporters gene) and MED (55 ABC transporters gene) (*Chen et al., 2016*; *Tian et al., 2017*). Subsequently, a comparative genomics analysis was performed between resistant and susceptible lines of MED biotype BtWf and 16 ABC transporters linked as insecticide resistant markers were identified (*Wang et al., 2023*). Nevertheless, another study on MED biotype BtWf also demonstrated that a substantial 80% of the ABCG subfamily exhibited significant upregulation in response to imidacloprid exposure. Hence, this finding clearly showed that ABCGs may contribute to the adaptability and resistance of BtWf to insecticides. To gain insight into these findings, *He et al. (2019)* performed a comparative analysis of the differential expression level of four ABCG subfamily genes in the MED biotype BtWf when subjected to imidacloprid. Among

these genes, *ABCG3* demonstrated the highest expression level in adult females after 6 h of exposure to imidacloprid.

In addition to metabolic resistance, target-site resistance is another factor contributing to insecticide resistance that can be revealed through omics analyses. As example, amplicon sequencing has been performed by comparing five different BtWf populations from various origins to determine the frequencies of *L925I* and *T929V* mutations in *para-type* voltage-gated sodium channels (VGSC) gene which are associated with pyrethroid resistance of BtWf (*Wei et al., 2021*). Pyrethroid toxicity is mediated by the closure of VGSC in the insect's axonal membrane, which causes the current to be prolonged during an action potential and disrupts the insect's electrical signals in their nervous system (*Pfeil, 2014*; *Ranathunge et al., 2021*; *Suppiramaniam et al., 2010*). *L925I* and *T929V* mutations in voltage VGSC may reduce or hinder the binding affinity for the insecticide, therefore negating the effect of insecticide in the BtWf (*Khan et al., 2020*; *Wei et al., 2021*). Similar procedure also was performed by *Lueke et al. (2020),* by utilizing sequenced RNA from five selected strains of MEAM1 and MED biotype BtWf with different resistance levels to identify the *A2083V* mutation in the ACC gene. This differences in the resistance level helps to determine common mutations, such as the substitution of specific amino acids in the ACC enzyme, which consequently involved in the resistance of keto-enol insecticides in BtWf.

Henceforth, genomics and transcriptomics analyses are able to provide holistic knowledge of insecticide resistance components in BtWf. The study of BtWf whole genetic material and its gene expression could identify resistance-related genes linked to the detoxification system or conferring target site insensitivity. As a result, these genes can be used as targets for the development of novel pesticides or the modification of existing ones, as well as for the discovery of potential RNA interference (RNAi) and CRISPR (Clustered Regularly Interspaced Short Palindromic Repeats) based genome editing targets (*Suhag et al., 2021*; *Zahoor et al., 2021*). These advanced technologies could be applied in current research to reduce the ability of BtWf to resist insecticides.

## Proteomics and metabolomics

Proteins and metabolites serve as a crucial link between genotype and phenotype, emphasizing the importance of understanding mechanisms and complex biological phenomena (*Aebersold & Mann, 2003*; *Fiehn, 2002*). However, their complexity and dynamic nature pose challenges (*Lakrisenko & Weindl, 2021*; *Miller & Smith, 2023*). Therefore, grasping the complexity of the proteome and metabolome is essential for advancing cellular biology and investigating insecticide resistance in BtWf. Proteomics integrates the discovery of protein sequence, expression, structure, and interactions, aiming for high throughput and reduced bias analysis (*Aizat & Hassan, 2018*; *Van-An & Lee, 2023*), while metabolomics involves a comprehensive analysis of metabolites or low-molecular-weight compounds in biological systems such as cells or tissues (*Oh et al., 2023*; *Pérez-Alonso, Carrasco-Loba & Pollmann, 2018*). Both analyses systematically identifies and quantifies these components involved in biochemical interactions, providing insight

into complex biological interactions, responses, and functions (*Fiehn, 2002*; *Oh et al., 2023*).

Most recent comprehensive proteomics analysis on BtWf involves both gel-based and gel-free techniques, such as 2-D gel electrophoresis (2DGE) and mass spectrometry-driven methods (*Coorssen, 2022*; *Kang et al., 2012*; *Mishra et al., 2016*). By comparing different variants and susceptibilities of BtWf, proteome changes may shed some light onto the resistance mechanisms of this pest (Table 1). For instance, *Kang et al. (2012)* conducted a comparative proteomics analysis integrating 2DGE with mass spectrometry (MS) to examine protein profiles between the two most destructive and invasive BtWf biotypes, MED and MEAM1. They identified several protein spots specific to the MEAM1 biotype, with carboxylesterase 2 (COE2), a Phase I detoxification enzyme, showing significantly higher expression than in the MED biotype BtWf. This suggests that the elevated COE2 expression in the MEAM1 biotype potentially offer superior defence against pyrethroid, and organophosphate insecticides compared to the MED biotype BtWf. Moreover, a comparative proteomics analysis of thiamethoxam-resistant and susceptible MEAM1 biotype BtWf have identified 52 differentially expressed proteins, with 38 proteins being upregulated (*Yang et al., 2013b*). Notably, there was increased expression of Phase I and Phase II detoxification enzymes, such as CYPs, UGTs, and GSTs, suggesting a metabolic detoxification mechanism associated with thiamethoxam resistance.

Furthermore, *Mishra et al. (2016)* also had utilized 2DGE but coupled with tandem mass spectrometry (MALDI TOF/TOF), and only analysed on MEAM1 biotype BtWf. They successfully profiled and identified 246 non-redundant proteins, leading to the selection of 11 sequences representing functional proteins of BtWf. These proteins were related to energy production, protein metabolism and stress response, which includes the Phase I and Phase II detoxification systems, such as the COE and GST (*Mishra et al., 2016*). A transporter protein was identified (ferritin), but it was not the ABC, the Phase III detoxification system transporter. Despite recent advances in mass spectrometry proteomic techniques, quantifying large and low-abundance transmembrane proteins, such as ABC transporters, remains challenging (*Al-Majdoub et al., 2020*).

Mass spectrometry (MS) and proton nuclear magnetic resonance (H-NMR) remains the most mainstream analytical platforms in metabolomics (*Zhao et al., 2024*). Metabolomic analysis can detects a wide range of unknown metabolites with diverse properties, enabling the screening of differential metabolites across treatments and the potential discovery of novel biomarkers (*Baharum & Azizan, 2018*; *Chen, Zhong & Zhu, 2020*; *Li et al., 2022*; *Resurreccion & Fong, 2022*). Therefore, ento-metabolomics approach was introduced to uncover biochemical insights, improve the understanding on insect physiology and behaviour (*Snart, Hardy & Barrett, 2015*). Despite the increasing number of insect's genomics, transcriptomics, and proteomics studies, the research on differential insects metabolism remains scarce, with fewer than 50 publications in the last decade (*Snart, Hardy & Barrett, 2015*). Nonetheless, there has been no research using metabolomics approaches on BtWf insecticide resistance activity to date.

Henceforward, the metabolomics approach is highly beneficial for uncovering detoxification mechanisms and pathways in BtWf. For instance, using an untargeted metabolomics approach (GC-MS) on another sap-sucking pest, *Aphis gossypii* exposed to varying imidacloprid dosages revealed adaptive insecticide resistance mechanisms, identifying 16 potential biomarkers linked to glycolysis, gluconeogenesis, and amino acid and antioxidant metabolism pathways. These findings highlighted high energy consumption, excitotoxicity, oxidative stress, and Parkinson's disease markers at higher concentrations (*Lv et al., 2021*). Similarly, lipidomics, which analyzes all lipids (*Swinnen & Dehairs, 2022*), combined with non-targeted metabolomics (UHPLC Q-Exactive Orbitrap MS) showed that glycerophospholipid, sphingolipid, and glutathione metabolism were elevated in Neuro-2a cells exposed to imidacloprid and acetamiprid (*Wang et al., 2021*). This increase is attributed to glutathione's role in xenobiotic metabolism and Phase II detoxification, facilitated by GST enzyme activity (*Tsuchida, 2002*).

Thus, through the lens of proteomics and metabolomics, the relationship between genotype and phenotype becomes distinctly comprehensible. The identification of enzymatic and metabolic pathways offers deeper insights into the chronological sequence of biological systems and their responses to insecticides or xenobiotics. Despite being underexploited, proteomics and metabolomics play a crucial role in augmenting genomic and transcriptomic data to achieve a comprehensive understanding of insecticide resistance mechanisms in BtWf. Integrating multi-omics platforms can expand our comprehension of intricate interactions and pathways, thereby enabling the development of more effective strategies for pest management and control.

## STRATEGIES IN COMBATTING BtWf

The application of insecticides does not appear to be a perfect solution for eradicating BtWf infestation as previous studies have shown its ability to resist insecticides. Auspiciously, multi-omics analysis provides a new paradigm to understand the mode of action for this resistance in the bigger picture. Henceforth, one of the proposed strategies to manage the BtWf infestation is by inhibiting their ability to detoxify and neutralize the xenobiotics. The multi-omics analyses have identified several highly expressed genes and proteins responsible for detoxification activity against various insecticides, such as abamectin, cyantraniliprole, pymetrozine, imidacloprid, chlorpyrifos, bifenthrin, and thiamethoxam (*Guo et al., 2020a*; *Wang et al., 2020a*; *Yang et al., 2013c*).

Initially, gene silencing systems were used as a tool to functionally determine and validate heterologous expressions and correlations of multi-omics hypothesized findings of detoxifying genes and enzymes from resistance BtWf (*Homem & Davies, 2018*). But as the gene silencing technologies evolved, advanced molecular editing procedures such as RNA interference-(RNAi) and CRISPR-(Clustered Regularly Interspaced Short Palindromic Repeats) based gene editing played a critical and beneficial role in knocking down the genes involved in xenobiotics detoxification and resistance to control such pests

(*He et al., 2019*, *2018*; *Seman-Kamarulzaman et al., 2023*; *Suhag et al., 2021*; *Zahoor et al., 2021*).

The success of the RNAi technique relies on delivering dsRNA into insects, which can be accomplished through methods such as microinjection, oral feeding, egg soaking, transfection, topical application, spraying with nanoparticles, leaf-mediated feeding, or using transgenic plants (*Karthigai Devi, Banta & Jindal, 2023*). For instance, *Upadhyay et al. (2011)* showed that RNAi is feasible to be applied in BtWf by feeding the pest with diets containing synthesized dsRNA and siRNA. As a result, they hypothesized that plant-mediated RNAi technology might be beneficial for controlling phloem-feeding pests, especially BtWf. Then, *Eakteiman et al. (2018)* effectively implemented plant-mediated RNAi on BtWf by downregulating the GST gene (*BtGSTs5*). These reports show that pathways that are heavily engaged in the breakdown of endogenous and foreign substances could be disrupted and suppressed for the BtWf control.

Nonetheless, delivering an effective lethal dose of dsRNA is a challenge for this strategy, which is due to the instability of unprotected dsRNA and the restricted dsRNA movement throughout the plants especially at the field level (*Suhag et al., 2021*). Thus, *Jain et al. (2022)* have improved the foliar spray-induced gene silencing (SIGS) approach by encapsulating dsRNA within a layered double hydroxide (LDH), known as BioClay. This innovative approach provides remarkable stability in environmental conditions and proves to be highly effective in selectively triggering RNAi to disrupt the entire life cycle of BtWf.

Apart from RNAi, CRISPR technology also has emerged to be a potent solution for overcoming the challenges posed by insecticide resistance among pests. By incorporating findings from multi-omics analysis, CRISPR has the potential to enable precise and targeted modifications of the resistance-associated genes, with the prospect of reinstating susceptibility to insecticides among pests (*Zahoor et al., 2021*). However, the use of this gene-editing technique is still relatively new and continues to undergo development and testing, particularly in the context of BtWf insecticide resistance analysis.

In regards to BtWf, CRISPR genome editing has proven valuable in identifying a novel mutation in the *acetyl-CoA carboxylase (ACC)* gene that is linked with a high level of resistance to keto-enol insecticides (*Lueke et al., 2020*). Ketoenols function as insect growth regulators by targeting the *ACC*, which is encoded by a single gene in insects, and catalyse the first crucial step in fatty acid biosynthesis (*Guest, Kriek & Flemming, 2020*). The mutation (A2083V) was identified by utilizing five BtWf strains with varying resistance profiles and geographical locations, which subsequently validated *via* CRISPR-Cas 9 genome editing in *Drosophila melanogaster*. The substitution of alanine with valine at position 2,083 alters the enzyme's function, contributing to the insecticide resistance phenotype (*Lueke et al., 2020*). Unfortunately, the genome editing validation was not performed directly in BtWf. One of the primary challenges in implementing gene editing in BtWf lies in the small size of their embryos, which makes direct injection challenging and often results in high mortality post-injection (*Heu et al., 2020*). Therefore, *Heu et al. (2020)* addressed this challenge by developing a CRISPR-Cas9 gene editing technique known as "ReMOT Control". This procedure involves the injection of a CRISPR-Cas9 ribonucleoprotein complex directly into the ovaries of vitellogenic adult females, rather

than into embryos. This approach was proven to be efficient and results in heritable gene editing in the offspring.

Additionally, CRISPR technology has advanced significantly in the model insect *Drosophila melanogaster*, enabling precise genome manipulation. For instance, *Fusetto et al. (2017)* successfully used CRISPR to knock down CYP6G1, a key enzyme in metabolizing xenobiotics like imidacloprid. *Kaduskar et al. (2022)* employed CRISPR to compare insecticide susceptibility by knocking down resistant (kdr) alleles in the voltage-gated sodium channels (VGSC) in *D. melanogaster*. Their study found that L1014F mutants were highly resistant to permethrin but susceptible to deltamethrin, I1011M mutants had moderate resistance to DDT and permethrin, and I1011V mutants remained susceptible to low concentrations of all insecticides. *Kaduskar et al. (2022)* also developed an allelic-drive system that reversed insecticide resistance linked to the L1014F mutation, restoring susceptibility over generations. This highlights the potential of allelic-drive technology to combat insecticide resistance. Beyond *D. melanogaster*, CRISPR has also been applied to other pest species like *Spodoptera, Helicoverpa, Tribolium, and Agrotis* (*Moon et al., 2022*).

Henceforth, this breakthrough opens the door to a deeper understanding of BtWf biology and offers the potential for novel solutions to insecticide resistance mechanisms. The ability to silence and modify the genome of BtWf provides means to identify genetic targets for the development of more efficient and targeted control strategies. Ultimately, this advancement may lead to more sustainable approaches for managing BtWf populations.

## CONCLUSION AND FUTURE PROSPECTS

This review details the use of systems biology omics platforms to elucidate the insecticide resistance mechanism in BtWf. Genomics and transcriptomics have proven to be the most on-demand platforms in current research due to their comprehensive, extensive data coverage and in-depth downstream analysis capabilities, particularly in the context of insecticide resistance. Despite being underutilized, proteomics and metabolomics would be advantageous and could offer crucial insights and validations into the enzymes and metabolic pathways that were closely associated with insecticide resistance in the future. Nonetheless, these omics studies highlight the key elements (genes, proteins and metabolites) of insect detoxification system, starting with Phase I detoxification of xenobiotics involving CYPs and COEs, followed by Phase II conjugation by UGTs and GSTs, and concluding with Phase III excretion of processed xenobiotics *via* ABC transporters. Furthermore, this integrated approach underscores the potential to discover novel insecticide resistance mechanisms beyond the detoxification system. They can uncover previously overlooked factors contributing to insecticide resistance in BtWf, such as target site resistance (including mutated genes or proteins) and inhibitors (metabolites). Accordingly, this lays the foundation for future research to pinpoint specific key elements for targeted analyses, such as RNAi and CRISPR genome editing, aimed at silencing or disrupting detoxification genes and critical enzymes involved in resistance mechanisms.

Therefore, focusing on these strategies holds promising potential for managing BtWf infestations effectively.

### Funding
The authors were supported by the Ministry of Higher Education (MoHe), Malaysia for the Fundamental Research Grant Scheme (FRGS/1/2020/STG01/UKM/02/9). The funders had no role in study design, data collection and analysis, decision to publish, or preparation of the manuscript.

### Grant Disclosures
The following grant information was disclosed by the authors:
Ministry of Higher Education (MoHe).
Malaysia for the Fundamental Research Grant Scheme: FRGS/1/2020/STG01/UKM/02/9.

### Competing Interests
The authors declare that they have no competing interests.

### Author Contributions
- Muhammad Aqil Fitri Rosli conceived and designed the experiments, performed the experiments, analyzed the data, prepared figures and/or tables, authored or reviewed drafts of the article, and approved the final draft.
- Sharifah Nabihah Syed Jaafar conceived and designed the experiments, analyzed the data, prepared figures and/or tables, authored or reviewed drafts of the article, and approved the final draft.
- Kamalrul Azlan Azizan conceived and designed the experiments, analyzed the data, prepared figures and/or tables, authored or reviewed drafts of the article, and approved the final draft.
- Salmah Yaakop conceived and designed the experiments, analyzed the data, prepared figures and/or tables, authored or reviewed drafts of the article, and approved the final draft.
- Wan Mohd Aizat conceived and designed the experiments, performed the experiments, analyzed the data, prepared figures and/or tables, authored or reviewed drafts of the article, and approved the final draft.

### Data Availability
This is a literature review.

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
