# Peer review of "Omics approaches to unravel insecticide resistance mechanism in Bemisia tabaci (Gennadius) (Hemiptera: Aleyrodidae)"

_PeerJ, doi:10.7717/peerj.17843_

## Round 0.1 · original submission · Minor Revisions

The authors are requested to revise the manuscript according to the suggestions of all three reviewers. The manuscript requires English editing.

**Language Note:** The Academic Editor has identified that the English language must be improved. PeerJ can provide language editing services - please contact us at [email protected] for pricing (be sure to provide your manuscript number and title). Alternatively, you should make your own arrangements to improve the language quality and provide details in your response letter. – PeerJ Staff

Reviewer 1 ·

Basic reporting

The review article "Omics approaches to unravel insecticide resistance mechanism in Bemisia
tabaci (Gennadius) (Hemiptera: Aleyrodidae) " (#92511) discussed omics techniques (genomics,
transcriptomics, proteomics, and metabolomics) in the study of insecticide resistance of B. tabaci.
Some major issues should be well clarified in order to improve the paper.

Major comments

Large amount of information of references were missing. And the uniform of reference should be
well checked according to the journal requirements.

Line 107- 128 the part of Mechanism of insecticide resistance should be rewritten. This part of the
discussion was not specific and lacked depth, simply listing some cases. Plenty of latest references
could be cited.

In addition, COE is Phase Ⅰ not Phase Ⅱ enzyme.

Line 132 some important references were not cited, for example, there were some references on
the genomics of B. tabaci, while there was only one reference (Chen et al., 2016). Readers
anticipate obtaining more valuable insights from the review article. Therefore, the author should
delve into the literature to uncover additional information and provide thoughtful analysis, rather
than simply presenting a series of cases.

Line 194 – 221 although three references on the protemics of B. tabaci were listed in the table 1,
none of them were stated, please find more information of these references, add your own
deliberation, and rewrite this part.

Line 223 – 227 please clear the difference between metabolomics and metabolism experiments.
For example, the references (Hamada et al. 2019; Jones et al. 2011; Nauen et al. 2013; Nauen et
al. 2015) were all about metabolism experiments.

Line 231 – 233 only one reference on the metabolomics (Wang et al., 2021) was cited in this part,
however, this reference was not relevant to the resistance of B. tabaci. Please find more references
on the metabolomics of B. tabaci and rewritten this part.

General comments

Line 113, 118, and 122 it should be Phase Ⅰ, Ⅱ, and Ⅲ, but not Phase 1, 2, and 3.

Line 154 – 157 this reference was not relevant to the content.

Line 159 – 160 Please remove this reference (Qin et al., 2023). The claim of discovering 24 novel
P450 genes in the article is incorrect, which would mislead readers.

Line 240 – 242 Please add a relevant reference to this statement.

Experimental design

.

Validity of the findings

.

Reviewer 2 ·

Basic reporting

The manuscript of “Omisc approaches to unravel insecticide resistance mechanism in Bemisia tabaci” collected some references to discuss the use of various omics to unravel the mechanism of insecticide resistance in whitefly Bemisia tabaci. Overall, most of information of this field in this manuscript is outmoded, many new references were published in these two years, reload these new results in the revision version.
Many insecticide treatment research results were used to verified the mechanism of insecticide RESISTANCE. This is not suitable. As all know, stable over-expression of detoxification enzyme genes or target mutations are the major mechanisms in the medium or high resistance populations. Some genes expression change after insecticides treatment is not traditional resistance mechanisms, recorrect many of these opinion though this manuscript.

Experimental design

this is a review manuscript

Validity of the findings

this is a review manuscript

Additional comments

NA

Reviewer 3 ·

Basic reporting

Please carefully read it once more, revise the introduction section, and make an effort to tie it to the main goal of the review. Cite recent papers as well.

This review paper requires more references related to omics approaches used in insecticide resistance mechanism in whitefly

Author should be included figure related to present status of insecticide resistance to different classes of insecticides in whiteflies.

Author should be included systemic figures of all omics approaches used in whitefly.

Author should be included list of enzymes used in proteomic approach in tabular form.

Author should be included list of differentially express proteins involved in detoxification of targeted insecticides in tabular form.

Author must be elaborate lipidomic, metabolomic approaches in detail with citation of latest references.

yes, review come within the scope of journal

Yes, this review paper illustrated a wide knowledge on numerous omic approaches which unrevealed the insecticide resistance.

Yes, introduction properly formulated but need to include some more latest references

Experimental design

Yes within scope

No comment

No comment on methods

Yes, survey methodology coverage almost of the subjects but need to include some figures and tables as as comments made in basic reporting

Yes mostly,

Yes, organized logically

Validity of the findings

no comment

conclusion and future prospects must be written again to support the review paper.

Yes, mostly set out in the introduction part.

conclusion and future prospects must be written again to support the review paper.

Additional comments

Author must be included more latest references related to RNAi and CRISPR technology to strengthen the review paper.

Author must be elaborated the genomic (DNA sequencing, genetic profiling, genetic mapping and structural and functional analyses of genome) transcriptomic (splicing and transcriptional regulation), role of epigenetics and lipidomic approaches in insecticide resistance in details .
The author should check whether the citation and bibliography are as per the format of the journal.

---

## Round 0.2 · accepted · Accept

I can confirm that the authors have addressed all of the reviewers' comments. I have reviewed and found the current version of the manuscript in its publishable form. Hence, I recommend its publication.